# Study on the Preparation and Properties of Jute Microcrystalline Cellulose Membrane

**DOI:** 10.3390/molecules28041783

**Published:** 2023-02-13

**Authors:** Zhengyong Liang, Xing Li, Meng Li, Yulu Hong

**Affiliations:** Department of Chemical Engineering, Zhengzhou University, Zhengzhou 450001, China

**Keywords:** jute, microcrystalline cellulose, cellulose membrane, microwave-assisted hydrolysis, separation

## Abstract

The preparation and performance control of the cellulose membrane are one of the hot topics in the environmentally friendly separation membrane field. In this study, microcrystalline cellulose (MCC) was prepared by microwave-assisted acidic hydrolysis of cellulose obtained from jute, followed by the use of a mixture of *N*-methylmorpholine-*N*-oxide and water as a solvent to obtain the homogeneous casting liquid, which was scraped and subsequently immersed in the coagulation bath to form a smooth and dense cellulose membrane. During membrane formation, the crystal structure of MCC changed from type I to type II, but the chemical structure remained unchanged. The mechanical strength and separation performance of the membrane were related to the content of MCC in the casting liquid. When the content of MCC was about 7%, the tensile strength of the membrane reached a maximum value of 13.49 MPa, and the corresponding elongation at break was 68.12%. The water flux (*J*) and rejection rate (*R*) for the bovine serum albumin were 19.51 L/(m^2^·h) and 95.37%, respectively, under an optimized pressure of 0.2 MPa. In addition, the coagulation bath had a significant effect on the membrane separation performance, and J and R were positively and negatively correlated with the polarity of the coagulation bath. Among them, it was note-worthy that J and R of membrane formed in ethanol were 33.95 L/(m^2^·h) and 91.43%, separately. Compared with water as a coagulation bath, J was increased by 74% at the situation and R was roughly equivalent, showing better separation performance. More importantly, the relationship between the structure and separation performances has also been studied preliminarily. This work provides certain guidance for the preparation of high-performance MCC membranes.

## 1. Introduction

In recent years, resource and environment issues arising from the nonrenewable and nonbiodegradable nature of fossil-based polymer materials have become more and more prominent, forcing the continuous exploration of new materials. Under this background, cellulose, as the most abundant and renewable natural resource in the world, has played a significant role [1,2,3,4]. Meanwhile, the rapid development of functional polymer membranes has strongly stimulated the social demand for high-strength, biodegradable, and special performance materials [5]. Relevant studies showed the regenerated cellulose membranes have the characteristics of strong regeneration, biodegradability, good biocompatibility, and thermochemical stability [6,7,8,9], and they have been used in the fields of packaging, agriculture, energy storage, electronic, pharmacy, and wastewater treatment, showing a huge application potential [10,11,12]. In fact, the inherent properties of cellulose membranes, such as hydrophilicity, water stability, sustainability, and excellent mechanical properties, make them more suitable for separation in the liquid phase [13,14]. Recent studies have demonstrated applications of cellulose membranes in water filtration, such as for oil/water separation, dye removal, or heavy metal adsorption [15,16,17].

As we know, the application direction and actual value mainly depend on the raw material and the membrane preparation process. At present, some progress has been made in cellulose membrane materials. Among them, one of the more prominent was MCC. MCC is a deep processed product of natural cellulose that is usually obtained by hydrolyzing cellulose for limiting the polymerization degree (LODP: 15-375) [18]. Its appearance is white, or nearly white, and consists of odorless and tasteless particles (20–80 μm) with light weight [19], high strength, fibrous, crystalline [20], and biodegradability [21] characteristics. Compared to conventional cellulose, the larger specific surface area and stronger intermolecular hydrogen bonding force make it more suitable as a membrane material. The preparation methods of MCC mainly include acid hydrolysis [22,23], alkaline hydrolysis [24], enzymatic hydrolysis, and combined methods [25], etc. Among them, the acid hydrolysis method has achieved widespread use, due to the advantages of mature technology, simple operation, and low operating costs.

Cellulose is difficult to process directly, due to its high melting point and crystallinity. Therefore, cellulose membranes are mainly prepared by a dissolution–regeneration method for their easy manipulation. In the dissolution process, cellulose molecules are tightly connected by a large number of hydrogen bonds, and the dissolution process is indeed difficult, so the choice of solvent is very critical. The physical and mechanical properties of viscose fibers were objectively well-established in traditional the viscose-based process, but the complex preparation process and the existence of exhaust gas pollution limited its further development. Therefore, the development of a new green cellulose solvent system has become an imperative task. So far, the new cellulose solvents that have been developed are *N*-methylmorpholine-*N*-oxide (NMMO) [26], ionic liquids [27], lithium chloride/dimethylacetamide (LiCl/DMAc) [28], alkali/urea or NaOH/sulfur urea, etc. [29,30]. Among these solvents, the advantages of NMMO system lie in its efficient physical dissolution process of cellulose, which is non-toxic and harmless [31], as well as easy to recycle and reuse, revealing a great development prospect.

Considering that jute is an abundant resource in the East and South Asia with high cellulose content, it was selected as the raw material in this work. During the process of cellulose extraction, the traditional strong alkaline NaOH solution cooking method with high temperature and high pressure [32,33] was abandoned, and we adopted a mild ultrasound-assisted Na_2_CO_3_-H_2_O_2_ solution extraction technology. Then, the cellulose was hydrolyzed with HCl (aq) under microwave radiation to avoid the shortage of a long time of conventional heating [34,35] and obtained high-quality MCC with a good yield. Finally, MCC was dissolved in NMMO-H_2_O system, and the phase exchange method was used to obtain the jute microcrystalline cellulose membrane. The chemical structure and thermal properties were evaluated by FTIR and TGA, respectively. The surface morphology, mechanical properties, and separation performance for bovine serum albumin (BSA) were characterized by XRD, SEM, and mechanical performance test. This work not only addresses the current situation of the single use and low economic value of jute, but also obtains a functional cellulose membrane with broad application prospects in the area of membrane separation.

## 2. Results and Discussion

### 2.1. Particle Size Analysis of Jute MCC

The hydrolysis degree of cellulose can be reflected by the size and distribution of the obtained MCC particles. The detailed particle size parameters of jute MCC and standard MCC are shown in Table 1. From Table 1, it can be found that the particle size of jute cellulose reached the micron scale, and the average particle size Dav is 19.16 μm. By comparison, the standard MCC had an average particle size Dav of 15.37 μm. The slight difference between them may be related to the type of raw materials and the manner in which they are treated. The particle size distribution of jute MCC (Figure 1) was relatively concentrated, which was near the distribution of standard MCC, indicating that the molecular scale of the prepared jute MCC is relatively uniform.

### 2.2. FTIR Analysis of Jute MCC Membrane

In the FTIR (Figure 2), the spectrum of jute cellulose and jute MCC were almost the same as the standard MCC before cellulose membrane formation. However, there were distinct differences after the membranes were formed. It is well-known that -OH, -CH, and -C=O stretching vibration peaks are at 3500–3200, 2895, and 1642 cm^−1^, respectively, which are the typical absorption peak of type I cellulose. After membrane formation, the -OH stretching vibration peak moved to 3295 cm^−1^, indicating that the hydroxyl group rearranged, and the C-H vibration peak moved from 2895 cm^−1^ to 2883 cm^−1^, indicating the regeneration occurrence of the redshift phenomenon. There was a weak absorption peak at 1739 cm^−1^, which is the C=O stretching vibration of acetyl and uronic acid ester groups in hemicellulose, indicating that a trace amount of hemicellulose remained, but this peak disappeared in the membrane, meaning the complete removal of hemicellulose during the course of membrane formation. Among them, the absorption peak at 1063 cm^−1^ became sharp, suggesting that the vibration peak of the C-O bond in the amorphous region became stronger after regeneration, as a result of the crystalline form’s change from type I to type II. At 848 cm^−1^, the amorphous band of the MCC membrane was obvious, indicating that the degree of amorphousness increased during the dissolution of MCC.

### 2.3. XRD Analysis of Jute MCC Membrane

As shown in Figure 3, the positions of the diffraction intensity corresponding to each crystal plane of the jute MCC and standard MCC were almost same, and the diffraction intensities both appeared at 2θ = 15°, 22.4° and 34.5°, indicating that the jute MCC maintained the natural cellulose-I type structure. After the dissolving and regeneration process, the MCC membrane showed an obvious broad diffraction intensity at 2θ = 20.3°, which indicated that the cellulose structure had transformed from cellulose-I to cellulose-II type [36]. According to the Segal formula, the crystallization indices of standard MCC, jute MCC, and MCC membrane were 79.05%, 72.93%, and 38.47%, respectively. The reason why the jute MCC crystallization index is slightly lower than that of the standard MCC may be related to the types of raw materials and the specific extraction and hydrolysis processes. There is a strong polar N-O bond in the NMMO molecule, which can easily form a hydrogen bond with -OH in cellulose, thus destroying the aggregation form of cellulose and making the cellulose dissolve in NMMO. Additionally, when the cellulose membrane was recrystallized in a coagulation bath, the cellulose molecules transformed into II type cellulose with a low crystallinity index. As seen in Figure 3, the diffraction peak of jute MCC membrane was wide and flat, which is the typical characteristic of the amorphous peak. The reason is that, when dissolved cellulose was regenerated from NMMO solution, the particle growth was not sufficient, and the molecules were not highly ordered [37]. As a result of this, the crystallinity index after membrane formation was significantly lower than that of cellulose.

### 2.4. SEM Analysis of Jute MCC Membrane

The surface structure of the jute MCC membrane is related to the MCC content in the casting liquid. Figure 4 shows the microstructure of the jute MCC membrane surface and cross-section. The surfaces of the membranes prepared with 5 wt% and 7 wt% of MCC were relatively smooth, but the surfaces of the membranes prepared with 9 wt% of MCC were relatively slightly rough, which may be caused by the excessive viscosity (38.0 Pa.s) of the casting liquid, and defects occurred easily during the scraping and drying process of the membrane, which could cause the surface states of the membranes to become worse. The cross-sectional structures of the membranes prepared with 5 wt% and 7 wt% of MCC were similar to sponge-like, uniform and without obvious defects. In contrast, the membranes prepared with 7 wt% of MCC were more compact. However, when the content of MCC increased by 9%, they were more likely to have defects. From the image of c’, some particles were prominent in the cross-section, which could be caused by the uneven arrangement of cellulose molecules, due to the shrinkage of the membrane during its drying process. Therefore, the solution prepared with 7 wt% of MCC was more suitable for membrane formation.

The effect of different coagulation baths on membrane morphology and structure has also been investigated. The membranes were formed in water, methanol, and ethanol as coagulation baths, respectively, with 7% casting liquid as the raw material. The surface and cross-sectional structures of the membrane samples were observed by SEM and were shown in Figure 5. It can be seen that a was flatter than b and c. In particular, b had obvious holes. At the same time, the cross-sectional view showed that the cross-section of the membrane was spongy, in which a’ was relatively smooth and flat, and b’ and c’ were relatively rough [38]. The reason is that, when the solidified casting liquid is immersed in the coagulation bath, the cellulose solvent molecules (NMMO) diffuse outward, and the coagulation bath molecules penetrate into the casting liquid simultaneously, in consequence, the double diffusion phenomenon between the coagulation bath and the solvent occurs. If the coagulation bath is methanol or ethanol, the speed of the coagulation bath entering the membrane is faster than the diffusion speed of the solvent molecules in the casting liquid. In this case, the cellulose macromolecules do not have enough time to rearrange and shrink, and as a result, the pore size left in the membrane is larger.

### 2.5. Thermal Stability Analysis of Jute MCC Membrane

To assess the feasibility of membrane applications at higher temperatures, thermal stability analysis was performed. As seen in Figure 6, at about 80 °C, the weight of the jute MCC and jute MCC membrane had a slight drop, indicating that they should release the same substance, which is a little absorbed water in the pores of the cellulose. Because cellulose is a hydrophilic substance, if it is not thoroughly dried, water will remain. Meanwhile, the rate of weight loss was the same for both jute MCC and jute MCC membrane, indicating the exchange between water and NMMO was relatively completed in the membrane forming process, and there was basically no residual NMMO in the MCC membrane. Two peaks appeared in the DTG graph, and the peak at about 229 °C represented the decomposition of plasticizer glycerol, and in the range of 310–340 °C, the decomposition of MCC contributed to weight loss. What is more, the initial pyrolysis temperature of jute MCC membrane was 170 °C, and the maximum weight loss rate temperature was 229 °C. By comparison, the initial pyrolysis temperature of jute MCC was 270 °C and the maximum weight loss rate temperature was 315 °C. The above results indicate that the thermal stability decreased after membrane formation, because the hydrogen bonds between cellulose molecules were gradually destroyed during the dissolution and regeneration process, and the intermolecular interaction force was weakened to some extent, affecting the thermal stability of the MCC membrane. In the carbon stabilization stage, the residues of the jute MCC and MCC membrane were 25.61% and 6.78%, respectively. The reason is that the membrane structure is loose, and its inside and outside are heated at the same time, which makes the membrane uniformly heated and easy to crack into volatiles. Therefore, the thermal stability is reduced, to some extent, after membrane formation.

### 2.6. Analysis of Mechanical Properties of Jute MCC Membrane

Figure 7 demonstrates the relations between the MCC content of the jute in the casting liquid and the mechanical properties of the membranes formed in water. As the MCC content increased, the tensile strength of the membrane firstly increased and then decreased slightly, from 5.88 MPa to the maximum value of 13.49 MPa, with the corresponding elongation at break reaching 68.12%. The reason is that the number of cellulose molecular chains per unit volume increases with the increase of MCC content, the density of the casting liquid, and the intertwining between molecules increase subsequently. Moreover, the molecular arrangement is more compact in space, which makes the order orientation high, and the total bond energy enhances. Consequently, the tensile strength and corresponding elongation at the break of the membranes gradually grows when the MCC content increases from 5% to 8%. However, when the content of MCC in casting liquid exceeds 8%, the concentration is too high objectively. The viscosity increases sharply, and structural defects are prone to occur in the process of scraping membrane, so the tensile strength and elongation at break also decrease to a certain extent.

### 2.7. Contact Angle Analysis of Jute MCC Membrane

The hydrophilicity or hydrophobicity of the membrane has some influence on its application fields. From the point of view of the structure of cellulose containing hydrophobic carbon rings and hydrophilic hydroxyl groups, it is clear that cellulose molecules should have certain amphiphilic property [39]. So, hydrophilic and hydrophobic properties of the membrane should be controllable in some ways. The contact angles of membranes formed by casting liquid with different cellulose contents were determined, and it was found that the contact angles of membranes-to-water were also different.

As can be seen from Figure 8, as the content of MCC in casting liquid increased from 4% to 9%, the contact angles of the membranes-to-water went from 21.30° to 55.93° correspondingly, which means that the hydrophilicity of the membrane decreased, resulting from the higher the content of MCC in the casting liquid and the denser membrane. Meanwhile, the shrinkage of the membrane pores resulted in the smaller specific surface area and less exposed hydroxyl groups, leading to a certain reduction of the hydrophilicity of the membrane.

### 2.8. Separation Performance of Jute MCC Membrane

The effect of MCC content in the casting liquid on the membrane flux and the rejection rate is shown in Figure 9. At the same pressure, the MCC content in the casting liquid was higher, the water flux was lower, and the corresponding membrane rejection rate was higher, due to the denser membrane and the smaller the pore size. At the same time, the MCC membrane water flux gradually increased with increasing pressure, but the growth rate gradually decreased. The reason is that the cellulose membrane is hydrophilic and easy to swell, which affects the growth rate of water flux. In general, the membrane water flux and rejection rate grew with the increase of pressure in positive and negative correlations, respectively.

Figure 10 shows the effect of different coagulation baths on the separation performance of membrane prepared under the same condition with 7 wt% of MCC. As the pressure increased, the fluxes gradually increased, and the BSA rejection rate gradually decreased simultaneously. It is known that the difference in the flux and rejection rate at the same pressure is related to the microstructure of the membrane. When the coagulation bath was methanol or ethanol, the cross-section of the membranes had finger-like pores left in the phase exchange process [40], and the pore size of membrane was obviously larger than that formed in the water coagulation. What is more, the flux of membranes formed in methanol, as the coagulation bath was greater than in ethanol as coagulation at the same pressure, which could be attribute to the polarity of coagulation bath: the polarity of the coagulation bath is smaller, and the pores left in the permeable membrane in the process of phase exchange are narrower, thus reducing the flux of the membrane. Although the flux of membranes formed in methanol was significantly larger than in ethanol, its rejection rate for BSA was obviously worse and may have lost practical value. Consequently, ethanol was the best choice as a coagulation bath from comprehensive consideration. As for the membranes formed in ethanol, the rejection rate was 91.43%, and the water flux reached 33.95 L/(m^2^·h) at 0.2 MPa. There was little difference in the rejection rate of the membrane formed in ethanol and water, and the water flux formed in ethanol was 1.74 times that formed in water. Therefore, in practical applications, depending on the nature of the material to be separated, as well as the operating conditions and cost, it is necessary to select a suitable coagulation bath, with the aim of obtaining a better flux and rejection rate.

## 3. Materials and Methods

Jute was produced in Bangladesh and purchased from Zhejiang Baisheng Industrial, Taizhou, China (cellulose content is about 67.59%). NMMO aqueous solution (50 wt%) and sodium hypochlorite solution (active chlorine content is 13%) were analytical pure and purchased from McLean, Palm Springs, CA, USA. Bovine serum albumin (BSA) was purchased from Solarbio, Beijing, China. Standard MCC, whose purity is over 99% and average diameter is less than 25 μm, was purchased from Aladdin Chemical Reagents company, Shanghai, China. Hydrogen peroxide (30%, *v*/*v*), nitric acid, anhydrous sodium carbonate, glacial acetic acid, anhydrous ethanol, anhydrous methanol, hydrochloric acid (37 wt%), propyl gallate and glycerol (98% purity) were purchased from Sinopharm Group, Beijing, China. All the regents had not been further purified before use.

### 3.1. Preparation of Jute MCC

A total of 5.00 g of jute, firstly, was boiled it in Na_2_CO_3_-H_2_O_2_ aqueous solution (12 wt%) for 3 h in a boron glass reactor accompanied by ultrasonic (300 W, 22 kHz), then filtrated, washed with water until neutrality, and dried to constant weight at 60 °C under vacuum, and the yield of the jute cellulose was about 92.36%. Secondly, crude cellulose was bleached with NaClO (aq)/CH_3_COOH solution (V_HAc_:V_NaClO_ = 1:1) at 100 °C for 1.5 h, washed, and dried again. Then, in a microwave reactor, the jute cellulose was hydrolyzed in 8% hydrochloric acid solution for 4–6 min under certain radiation intensity. After the time ran out, a certain amount of cold water was immediately added to terminate the reaction. The crude jute MCC was filtrated out and washed to neutrality, then fully dried to obtain 2.76–2.93 g MCC. The yields under different operation conditions were between 88.20% and 93.89%. 

### 3.2. Preparation of Jute MCC Membrane

NMMO-H_2_O solution (50 wt%) was distilled under reduced pressure to a water content of about 13.3%, then added to 0.5% propyl gallate and stored in a cool place away from light [41]. A certain amount of jute MCC was added to NMMO-H_2_O solution and swollen at 100 °C for 8 h. Then, it was stirred violently using a mechanical stirrer, until MCC was completely dissolved and formed an amber transparent casting liquid. Secondly, the casting liquid was kept at the same temperature for complete degassing, and membrane was scraped on a preheated glass plate at constant speed with an I-shaped coater (size of thickness is 250 μm). After scraping the membrane, put it in coagulation bath (defaults to water, unless otherwise specified) quickly for 24 h, and we changed bath liquid at intervals of 8 h. Finally, the completely solidified membrane was put into 30% glycerol aqueous solution and plasticized for 1 h, then let it air-dry and subsequently placed in a vacuum drying oven at 50 °C for drying to constant weight. Cellulose membrane was obtained with a thickness of about 13 μm, measured by a film thickness tester A3-SR-100.

### 3.3. Characterization

Particle size distribution of MCC was measured by a LA-960 laser particle size analyzer (Horiba, Kyoto, Japan), and the samples were dispersed in water to form a 1% suspension. A Fourier transform infrared spectrometer (Nicolet-iS5, Madison, Wisconsin, USA) was used to characterize the typical chemical groups of samples (MCC samples were embedded in KBr pellets, membrane samples were cut into long strips of about 2.0 cm × 5.0 cm, and glued to the glass sheet with double sided adhesive) in the range of 4000–400 cm^−1^. The contact angle was measured by JCY-4 contact angle measurement with dynamic contact angle measurement function (Fangrui, Shanghai, China).

#### 3.3.1. XRD Analysis

Firstly, the MCC sample was dried to constant weight at 80 ℃ in vacuum, and then milled into powder more than 300 mesh with an agate mortar under an infrared lamp. The XRD patterns of the samples were recorded on a D8 diffractometer (Bruker, Karlsruhe, Germany) using the graphite monochromatized Cu Kα radiation in the 2θ range 3–50° with step size of 0.02° under the operational conditions of 40 KV and 40 mA, scanning speed 10°/min. The wavelength was 0.15418 nm, and the analysis mode was adopted as normal reflection. The MCC membrane samples were usually more than 50 μm thick, and they can be cut into rectangles with certain size for XRD analysis directly. Based on the XRD results, the crystallinity index of cellulose (CrI) was calculated according to the literature [42]. The crystallinity index of cellulose membrane was calculated according to the literature [43]. 

#### 3.3.2. SEM Analysis

The surface topography of membrane sample was observed by JCM-6000 scanning electron microscope (JEOL, Tokyo, Japan). The dried jute MCC membrane was cut into appropriate size to make section, adhered to the carrier with conductive adhesive, and then scanned at the accelerating voltage of 10 KV.

#### 3.3.3. Thermal Gravimetric Analysis

TGA was carried with a TGA2 thermal analyzer (Mettler-Toledo, Greifensee, Switzerland). A total 5.0 mg of sample was put into an alumina crucible, which was heated from 25 ℃ to 500 ℃, with a heating rate of 10 ℃/min under a nitrogen atmosphere.

#### 3.3.4. Mechanical Performance Test

The membrane samples were cut into a rectangle of 2.0 cm × 6.0 cm. The mechanical properties were tested on a WDW-100 universal tensile testing machine (Guanteng, Guangzhou, China). The tensile speed was 50 mm/min, and the clamp was 50 mm. The testing method was referred to GB/T1040.3-2006 of China.

#### 3.3.5. Separation Performance Test

The membrane was cut into a circle with diameter of 4 cm and measured with a 1.00 g/L BSA aqueous solution, pre-pressured firstly at 0.2 MPa for 10 min to stabilize the system, then operated at the set pressure for 30 min, and the permeating liquid was collected and measured. The membrane permeating flux and rejection rate were calculated according to the following Equations (1) and (2), respectively, and the concentration of BSA in the permeate was measured by a UV-2401PC spectrophotometer (Shimadzu, Kyoto, Japan) and compared with the original solution to obtain the rejection rate.
(1)$$J=\frac{V}{St}$$
where *J* is flux of the membrane, L/(m^2^·h), S is the effective area of the membrane (m^2^), *t* is the test time (h), and *V* is the volume of permeating liquid in test time (L).
(2)$$R=(1-\frac{C_{1}}{C_{0}})\times 100\%$$
where *C*_0_ and *C*_1_ are concentration of BSA in the stock solution and permeation solution, respectively (g/L), and *R* is the rejection rate of the membrane.

## 4. Conclusions

In this work, MCC was prepared by microwave-assisted hydrochloric acid hydrolysis, with an excellent yield of 93.89%, using abundant jute as the material. The average particle size Dav of MCC was about 19.16 μm near to the standard MCC purchased from Aladdin Chemical Reagents company, Shanghai, China. Further, jute MCC membranes were successfully prepared by the dissolution–regeneration method. The dissolution of jute MCC in the NMMO-H_2_O solvent belongs to a physical process, and the cellulose crystal form changes from type I to type II after membrane regeneration. It was found from our research that the MCC content in the casting solution had a significant effect on the membranes’ properties, such as the aggregated structure, surface morphology, hydrophilicity, and mechanical properties. When the MCC content in the casting liquid was 7%, the obtained membranes had superior integrated properties, compared to the others. From the results of separating BSA, the rejection rate and water flux were positively and negatively correlated with the MCC content and the polarity of the coagulation bath, respectively. Under the optimal conditions of 0.2 MPa pressure and ethanol as coagulation bath, the flux and rejection rate for BSA were 33.95 L/(m^2^·h) and 91.43%, respectively. Compared with the membrane formed in water as a coagulation bath, the rejection rate had no significant difference, but the membrane flux was 1.74 times that formed in water. However, in view of the diversity of the separated objects in practical applications, it is necessary to choose a suitable balance between the rejection rate and the water flux to meet the requirements of a particular process. In general, the jute MCC membrane has excellent properties, whose structure can be controlled easily, and it may have a wide application prospect.

## Figures and Tables

**Figure 1 molecules-28-01783-f001:**
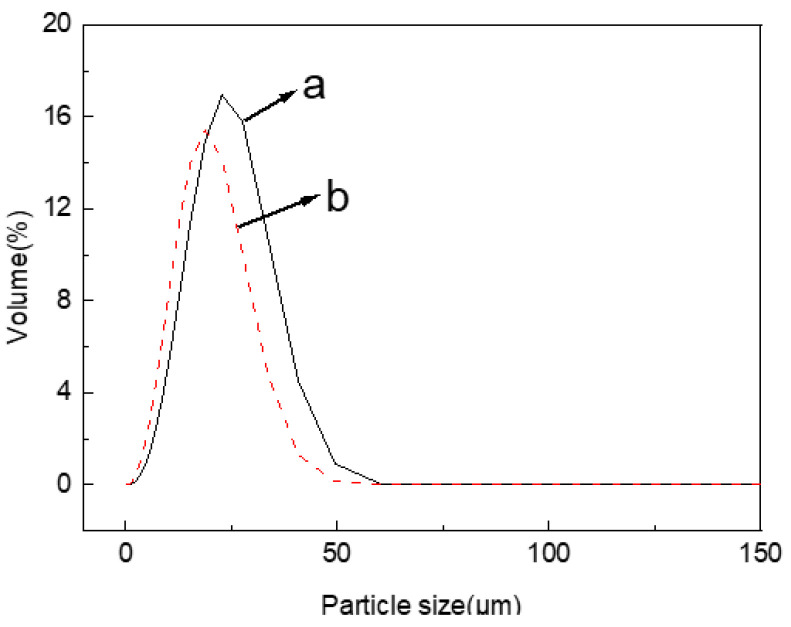
Particle size distribution of jute MCC (a), and standard MCC (b).

**Figure 2 molecules-28-01783-f002:**
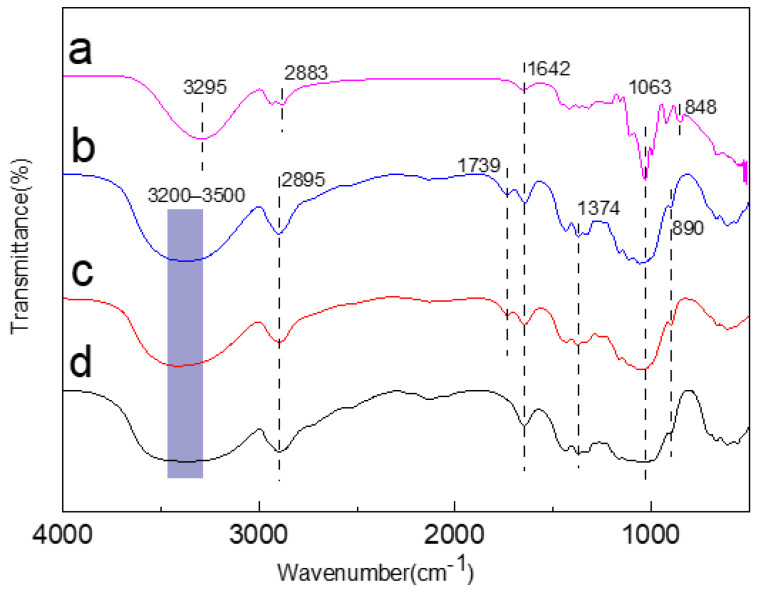
FTIR spectrum of jute MCC membrane (a), jute MCC (b), jute cellulose (c), and standard MCC (d).

**Figure 3 molecules-28-01783-f003:**
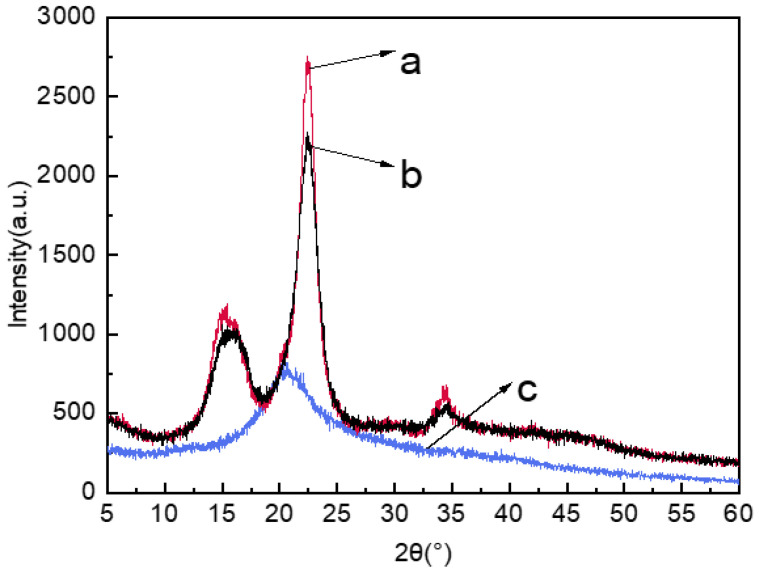
XRD images of standard MCC (a), jute MCC (b), and jute MCC membrane (c).

**Figure 4 molecules-28-01783-f004:**
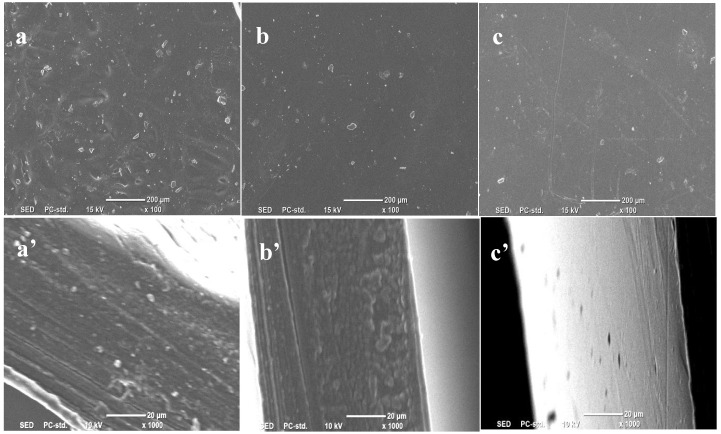
SEM images of jute MCC membranes prepared from casting solution with different jute MCC content: (**a**–**c**) the surface and (**a’**–**c’**) the cross-section. Note that Images (**a**), (**b**), and (**c**) represent the plan views of membranes formed by 5 wt%, 7 wt%, and 9 wt% of MCC, respectively; Images (**a’**), (**b’**), and (**c’**) represent the cross-sections of membranes formed by 5 wt%, 7 wt%, and 9 wt% of MCC, respectively.

**Figure 5 molecules-28-01783-f005:**
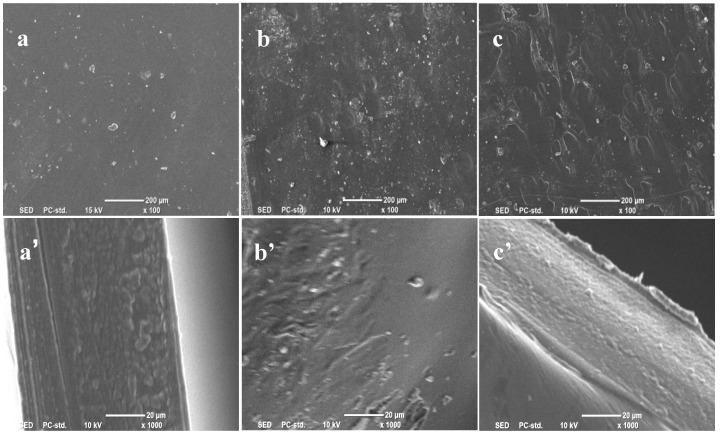
SEM images of jute MCC membranes formed in different coagulation baths: (**a**–**c**) the surface and (**a’**–**c’**) the cross-section. Note that (**a**) and (**a’**) are the plan view and cross-sectional view of the membrane formed in water coagulation bath, respectively; (**b**) and (**b’**) are the plan view and cross-sectional view of the membrane formed in methanol coagulation bath, respectively; (**c**) and (**c’**) are the plan view and cross-sectional view of the membrane formed in ethanol coagulation bath.

**Figure 6 molecules-28-01783-f006:**
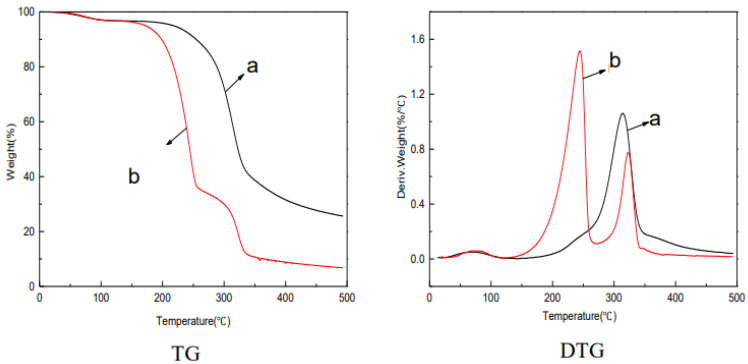
TG and DTG graphs: jute MCC (**a**) and jute MCC membrane formed in water as coagulation baths with 7 wt% of MCC (**b**).

**Figure 7 molecules-28-01783-f007:**
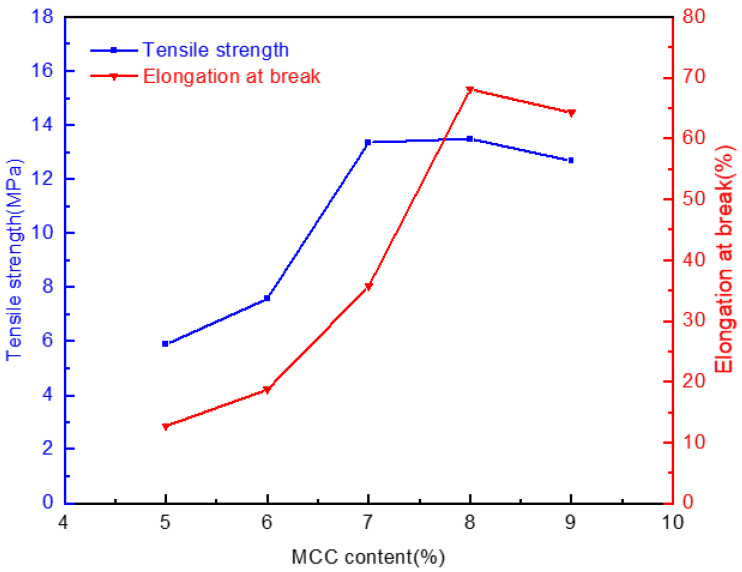
Effect of jute MCC content in casting liquid on the mechanical properties of membrane.

**Figure 8 molecules-28-01783-f008:**
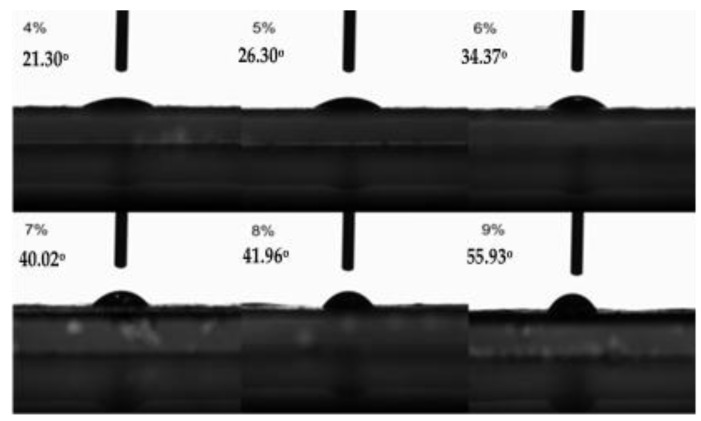
Contact angles of membranes prepared from casting liquid with different jute MCC content.

**Figure 9 molecules-28-01783-f009:**
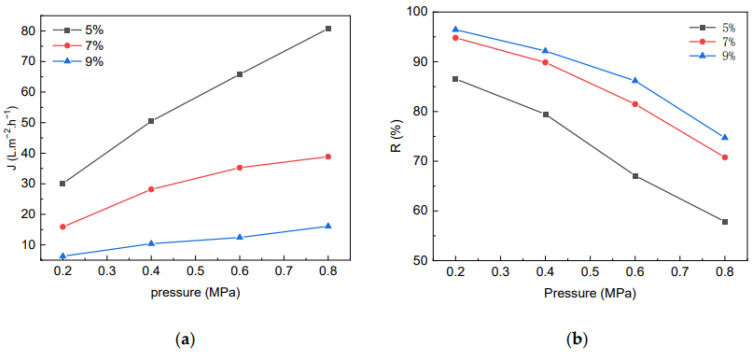
The effect of MCC content in casting liquid on Flux (**a**) of membrane and rejection rate for BSA (**b**).

**Figure 10 molecules-28-01783-f010:**
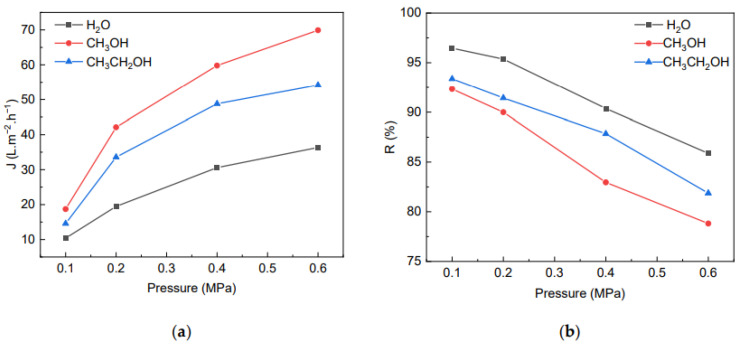
The effect of different coagulation bath on flux (**a**) of the membrane and rejection rate for BSA (**b**).

**Table 1 molecules-28-01783-t001:** Granularity parameter table.

Category	D_10_ (μm)	D_50_ (μm)	D_90_ (μm)	Dav (μm)
Prepared jute MCC	8.47	18.53	31.08	19.16
Standard MCC	6.15	14.66	25.83	15.37

## Data Availability

All data generated or analysed during this study are included in this published article.

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
