# Peer review of "Study on the Preparation and Properties of Jute Microcrystalline Cellulose Membrane"

_molecules, 2023, doi:10.3390/molecules28041783_

Round 1
Reviewer 1 Report
The authors reported a study dealing with the preparation and characterisation of jute microcrystalline cellulose MCC membrane. The idea is to develop MCC from jute and compare with commercially-available standard MCC. It’s a good paper.
The paper could be eligible for publication in Molecules after modifications. Some points need to be addressed:
1) Materials: Please add the origin of standard MCC.
2) Figure 3 is unavailable.
3) What do you mean by “M1” in Figure 6 ?
4) Please avoid to use MCC content in casting liquid because this material is a membrane. Please could you use membrane prepared with 9 wt% of MCC, for example.
5) Do you have measured the solution viscosity because some results are indicated as caused by excessive viscosity. Could you evaluate this viscosity?
6) In my opinion, the section dealing with thermal stability needs to be rewrite. The TG and DTG graphs of jute MCC and jute MCC membrane are different (line 212) because the thermal stability is decreased for the MCC membrane. Indeed, there is a weight loss at about 80°C which is undefined.
7) Please could you check the crystallinity degree of MCC by DSC. Any comment on thermal characteristics are reported. DSC measurements are required here.

Reviewer 2 Report
First of all, I consider that the article” Study on the Preparation and Properties of Jute Microcrystalline Cellulose Membrane” authored by Zhengyong Liang, Xing Li, Meng Li and Yulu Hong is not suitable for “Functional Carbon Quantum Dots: Synthesis and Applications” special issue. It not exist a correlation between issue topic and proposed article. Also, the article does not contain novel/important information in the field.
Observations
- Please check the language.
- Please check the text, there are many fragmented words
Introduction
- Please check the language.
Ex: The chemical structure, thermal performance, surface morphology, mechanical properties and separation performance for bovine serum albumin (BSA) were characterized by FTIR, XRD, SEM, TGA, and mechanical performance test.
The chemical structure and thermal performance are evaluated, are not characterized!
Results and discussion
2.2. FTIR analysis of jute MCC membrane
In the FTIR (figure 2), the spectrum of jute cellulose and jute MCC were almost the same as to the standard MCC before cellulose membrane formation- please re-phormulate!
2.3. XRD analysis of jute MCC membrane
There is a very polar N-O bond in NMMO molecule and -OH in cellulose is easy to form hydrogen bond with N-O bond, thus destroying the intermolecular force of cellulose and further destroying the aggregation form of cellulose. Therefore, in the process of cellulose dissolution, the crystallization zone was destroyed to some extent. Even though the cellulose membrane is recrystallized, the crystallinity index was still less than that of cellulose- please check and re-phormulate!
2.7. Contact angle analysis of jute MCC membrane
“The contact angle is an index to measure the hydrophilic or hydrophobic properties of the membrane. Hydrophilicity or hydrophobicity of the membrane have a certain influence on its application fields. Thus, it is great sig-nificant to control the hydrophilic and hydrophobic properties of the membrane. From perspective of cellulose structure containing hydrophobic carbon ring and hydrophilic hydroxyl group, it is clear that cellulose molecule should have certain amphiphilic property [39]. Therefore, hydrophilic and hydrophobic properties of the membrane should be controllable in some ways. The contact angles of membranes formed by casting liquid with different cel-lulose contents were determined, and found that the contact angles of membranes to water were also different” - this text is not absolutely necessary, it contains basic information known to readers.
Round 2
Reviewer 1 Report
The authors reported a study dealing with the preparation of jute microcrystalline cellulose MCC membrane and characterisation by means of FTIR and XRD analyses, SEM observation, thermal and mechanical characterization, and contact angle measurement.
The revised paper is improved. Some points need to be addressed:
1) Please evaluate the degree of crystallinity of membranes by using XRD analysis. Do you obtain a difference or not ?
2) Please use the right unit of viscosity: Pa.s
3) In my opinion, using M1, M2 and so on in the text do not help to follow the interpretation. Could you add a correct designation or a scheme?

Reviewer 2 Report
The authors made some corrections. I still consider that this article is not suitable for “Functional Carbon Quantum Dots: Synthesis and Applications” special issue. It not exist a correlation between issue topic and proposed article.
